# Personalized Beyond-accuracy Calibration in Recommendation

## ABSTRACT

Recommender systems usually aim to optimize accuracy in a supervised setting. Due to various data biases, they often fail to enhance other critical qualities that go beyond accuracy, such as diversity, novelty, and serendipity. Prior studies focus on addressing the bias in beyond-accuracy metrics from the provider's perspective, such as increasing the overall diversity of recommendations to combat popularity bias. In this work, we take a user-centric approach to this problem and demonstrate that users have distinct preferences for beyond-accuracy metrics. We hypothesize that users have an implicit behavioral model that goes beyond optimizing their choices only for accuracy. For instance, we assume that a user's purchase behavior is a mix of items that are more familiar to the user (optimizing for accuracy), and new items that are aimed for exploration (optimizing for novelty). We argue that a recommender system should reflect users' interest in such beyond-accuracy metrics. This perspective allows for a more holistic understanding of users' behavior and preferences leading to more fine-grained personalized recommendations. To this end, we propose a post-ranking greedy optimization algorithm that ensures recommendations are not only accurate but also meet users' beyond-accuracy preferences. Through extensive experiments, we demonstrate our proposed method's ability to balance the trade-off between ranking accuracy and user-centric beyond-accuracy preferences.

## CCS CONCEPTS

• **Information systems** → **Information retrieval**; **Recommender systems**.

## KEYWORDS

Calibration, Recommender Systems, Fairness, Re-ranking

**ACM Reference Format:**
Anonymous Author(s). 2023. Personalized Beyond-accuracy Calibration in Recommendation. In *Proceedings of the 10th ACM SIGIR / The 14th International Conference on the Theory of Information Retrieval (ICTIR '24), July 14-18, 2024, Washington D.C., USA.* ACM, New York, NY, USA, 10 pages. https://doi.org/10.1145/nnnnnnn.nnnnnnn

## 1 INTRODUCTION

Many systems deploy recommender systems (RSs) to assist users in finding the target relevant content among the vast array of options available. One of the most widely used classes of recommender algorithms is collaborative filtering. These algorithms often exploit the user's prior interactions to enhance the effectiveness of

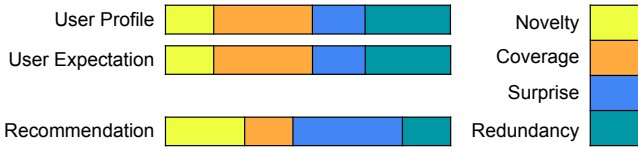

**Figure 1: Mismatch between user's performance expectations based on their prior interactions and the actual performance of the recommendation algorithm that may not be personalized to meet those expectations.**

the recommendations, measured by typical ranking-based metrics (e.g., nDCG, Recall). Recent literature [8, 11] suggests that such a methodology fails to account for crucial aspects of recommendation, including propagation of natural biases in data (e.g., popular items) that may significantly impact the overall quality of recommendations "*beyond accuracy*" (e.g., novelty, diversity) [6, 37]. For instance, such algorithms might end up with high accuracy metrics by recommending a small set of popular items while carrying the risk of gradually narrowing down the users' areas of interest, creating an effect similar to echo chambers [9].

The propagation of data biases in beyond-accuracy measurements is extensively studied from the perspective of item providers [39, 40, 43]. A recurring theme in the debiasing literature often focuses on a particular *criterion to group* items, framing the problem as a building algorithm that fulfills a recommendation *objective* within these groups. Grouping criteria are often either based on a demographic attribute (e.g., provider continent [15] or gender [5]) or an interaction-based attribute (e.g., popularity [3]). The objective could be goal oriented [27] or based on (weighted) parity among groups [25].

In this work, we focus on goal-oriented beyond-accuracy debiasing from the users' perspective; that is, users should receive recommendations tailored to their preferences and/or behavior regarding beyond-accuracy measurements. For instance, we see in Fig. 1 that a user who has interacted with a sufficiently large set of items in their profile (so excluding a cold-start user). Computing beyond-accuracy measures on the set of items in the user profile shows that the user has a preference for interacting with a wide range of categories multiple times (having high redundancy and catalog coverage [37]) rather than novel and surprising items. Accordingly, it is intuitive to expect that the beyond-accuracy measures proportionality remain the same in a personalized recommendation list [38]. Therefore, recommender systems should calibrate their recommendations to account for it. We refer to this task as personalized beyond-accuracy calibration (PBAC). A simplified use-case example of PBAC can be found in the next-basket recommendation literature [4, 24] where each user's recommended basket is divided into a repeat and exploration part. The division is inspired by user grocery shopping behavior that consists of regular weekly items (repeat), as well as new items that have not appeared in the user's basket before

(explore). We see this as a specific case of PBAC, already demonstrating the importance and significance of it in the next-basket recommendation domain.

In particular, we aim to answer the following research questions:

**RQ1** What is the extent of variations in user preferences for beyond-accuracy metrics and is it possible to effectively group users based on these preferences?

**RQ2** How well do widely used recommendation algorithms address beyond-accuracy variations in user preferences? Are there any user groups that receive favorable recommendations regarding beyond-accuracy calibration?

**RQ3** Can our proposed PBAC algorithm be used to determine the optimal balance between accuracy and beyond-accuracy calibration in a recommender system for different user groups?

**RQ4** To what extent does the underlying recommendation algorithm impact the effectiveness of our proposed PBAC re-ranking model?

In order to answer **RQ1** and **RQ2**, we utilize *K-means* clustering as a criterion to group similar users according to their propensity toward beyond-accuracy metrics. Our experiments on two real-world datasets with diverse domains showcase that users significantly differ in their preferences regarding beyond-accuracy metrics. However, state-of-the-art collaborative filtering techniques such as VAECF [26] and WMF [16] fail to account for this. We argue that incorporating beyond-accuracy calibration into the recommendation process is essential to ensure that the recommendations meet the diverse needs and expectations of all users. Recommender systems are often trained in a pairwise or pointwise manner, making it challenging to include calibration objectives in the training phase as it is a list-wise property [35, 42]. As a result, in response to **RQ3**, we propose a *re-ranking* mixed-integer optimization framework aiming at simultaneously optimizing for accuracy and multi-objective PBAC metrics to achieve the best trade-off between these objectives. We further propose a greedy algorithm for solving this problem in polynomial time and evaluate the performance of our algorithm for all clusters of users on Movielens1M and Yelp datasets (Experimental setup is outlined in Sec. 5). Results demonstrate our proposed method's effectiveness in satisfying PBAC property for all clusters of users, while maintaining the accuracy of recommendation. Moreover, to address **RQ4**, we compare the performance of several state-of-the-art recommendation algorithms in satisfying users' preferences in terms of beyond-accuracy metrics after applying the proposed model. Our key contributions, in summary, are as follows:

- We highlight the importance of beyond-accuracy calibration for users by analyzing their preferences and the performance of state-of-the-art collaborative filtering algorithms in representing these preferences.
- We propose a re-ranking optimization algorithm capable of simultaneously optimizing accuracy and beyond-accuracy calibration property in polynomial time.
- Through extensive experiments, we evaluate our algorithm on two real-world datasets and compare it against five baseline algorithms.

## 2 RELATED WORK AND CONCEPTS

**User-oriented algorithmic bias in recommendation.** *Algorithmic bias* is the bias induced or propagated by algorithms' mechanisms [8]. Algorithmic bias is often studied either from the user or item perspectives. User-oriented algorithmic bias is particularly concerning as it can reinforce stereotypes and discrimination based on various factors such as race, gender, and age [1, 12]. From the users' perspective, Ekstrand et al. [12] explore the disparities of accuracies within groups of users with different demographic characteristics and then propose a re-sampling method to address the issue. Wang and Chen [38] demonstrate a disparity in the distribution of beyond-accuracy metrics with respect to different user characteristics and highlight the need for algorithms capable of addressing it. Moreover, Li et al. [25] propose an optimization method to create parity between active and inactive user groups. Rahmani et al. [30] perform extensive experiments on the generalizability of the user-oriented algorithmic fairness method proposed by Li et al. [25] under different grouping assumptions and diverse domains. Their results indicate that grouping users based on a single attribute or a cutoff (e.g., top 5% of most active users) is domain-dependent with poor generalizability, requiring more nuanced and multidisciplinary approaches to algorithmic bias. Similarly, this paper examines the biases induced by algorithms from the user's perspective. Furthermore, we utilize K-means clustering to structurally divide similar users in their tendency regarding beyond-accuracy metrics into clusters by minimizing within-cluster distance. In contrast to the previous works on user-oriented algorithmic bias, this work aims to present each user with recommendations that match their expectations.

**Beyond-accuracy metrics in recommendation.** Researchers propose several beyond-accuracy metrics to evaluate the quality of a recommendation algorithm. Smyth and McClave [34] propose the average pairwise distance between items in a recommendation list to measure recommender system diversity. A similar metric is proposed by Ziegler et al. [44] to measure "intra-list similarity" scores in which high values denote low diversity. Another line of research in beyond-accuracy metrics focuses on improving such qualities in recommender systems. For instance, Kamishima et al. [20] and Abdollahpouri et al. [2] propose methods to enhance the long-tail coverage of the recommendation algorithm. Pardos and Jiang [29] investigate the filter bubble problem in a university course recommendation system and propose a method for bursting the bubble by recommending a more diverse set of items. This study utilizes the metrics defined by prior works to extract users' propensity towards these qualities. However, we optimize recommendation algorithms to meet users' expectations by calibrating the recommendation list accordingly.

**Calibration fairness.** There has recently been a renewed focus on the calibration concept in the context of fairness in machine learning and classification tasks [7]. Calibrated algorithms are those in which the predicted proportions of the various classes match the actual proportions of data points contained in the training data [13]. This ensures that the algorithm is not biased towards any one class, and that the predictions it makes are accurate and reflective of the actual data. In recommendation tasks, Steck [36]

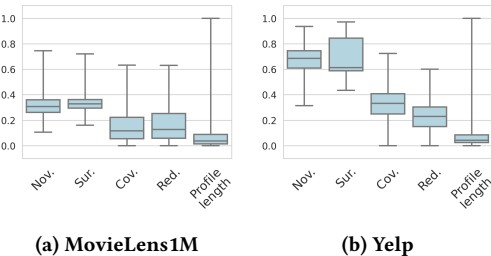

(a) MovieLens1M          (b) Yelp

**Figure 2: The distribution of extracted beyond-accuracy metrics on each dataset.**

proposes an iterative greedy algorithm for ensuring the distribution of recommended movie genres is consistent with the users' historical data. Rojas et al. [32] examine how the method proposed by Steck [35] performs when dealing with bias in several collaborative recommendation algorithms and the effect of genre calibration on accuracy. Klimashevskaia et al. [22] study the impact of three different re-ranking-based (post-processing) calibration approaches, analyzing how they can amplify or mitigate popularity bias in the movie recommendation domain. Ariannezhad et al. [4] apply the concept of calibration in the next-basket recommendation scenario, where the division of recommended baskets into repeat and exploration sections [24] is based proportionally on the historical interactions of the users in grocery shopping domain. Contrary to the works surveyed, this paper focuses on calibrating beyond-accuracy measurements in the recommendation list with the same proportion as users' profiles in training data.

## 3 BEYOND-ACCURACY CALIBRATION

In this section, we motivate the personalized beyond-accuracy calibration concepts by undertaking both data and algorithmic analyses. Firstly, we discuss the background and beyond-accuracy metrics, and then we analyze users' historical interactions to derive their tendency towards beyond-accuracy qualities. In addition, we group users according to their similar tendencies. Finally, we assess how well state-of-the-art recommendation algorithms can match the beyond-accuracy offered in their recommended list to the historical preferences of the users. This assessment allows us to determine the degree to which the algorithms are able to personalize recommendations to individual user preferences.

### 3.1 Beyond-accuracy metrics

Prior studies [14, 19] define several metrics for evaluating the quality of recommender systems that go beyond accuracy. These metrics are selected based on the application and objective of the recommender system. Following the most widely adopted beyond-accuracy metrics and preliminary correlation analysis, we select the metrics with the least correlation with one another to capture extractable characteristics. Hence, we choose four beyond-accuracy metrics along with profile length as grouping criteria. This is because user profile length plays a crucial role in determining the confidence of our estimates of tendencies toward beyond-accuracy metrics. Although our selection is not comprehensive, we believe these four metrics are representative of the full range of extractable

characteristics and are suitable for use in our experiments. The idea and analysis we present are trivially applicable to any arbitrary set of beyond-accuracy metrics.

(1) **Global long-tail novelty (Nov.)** measures the user's willingness to interact with unpopular items. The value ranges from 0 to infinity. Higher values indicate a higher tendency for the user to interact with unpopular items. Following [6], we define *novelty* as:

$$\text{Novelty}(\mathcal{P}) = \frac{-1}{|\mathcal{P}|} \sum_{i \in \mathcal{P}} \log_2(p(i)) , \qquad (1)$$

where $\mathcal{P}$ indicates the set of items in the user history and $p(i)$ is the popularity score of item $i$.

(2) **Co-occurrence-based surprise (Sur.)** measures the user's interest in interacting with unexpected items. Formally defined as:

$$S_{co-occ}(i, \mathcal{P}) = \frac{1}{|\mathcal{P}|} \sum_{j \in \mathcal{P}} PMI(i, j) , \qquad (2)$$

where $PMI(i, j)$ denotes the pairwise mutual information score between items $i$ and $j$, i.e., the extent to which observed co-occurrences differ from what is expected [18]. Given a pair of items $i$ and $j$, $PMI(i, j)$ is calculated as follows:

$$PMI(i, j) = \frac{\log_2(\frac{p(i,j)}{p(i)p(j)})}{-\log_2(p(i, j))} , \qquad (3)$$

where $p(i)$ and $p(j)$ indicate the probabilities that a user interacts with items $i$ and $j$, respectively, and $p(i, j)$ shows the probability that a user interacts with both items PMI values range from $-1$ to 1, with $-1$ indicating no user interacts with the two items together (negative co-occurrence) and 1 signifying complete co-occurrence of the two items. Therefore, $S_{co-occ}$ also takes values from $-1$ to 1. The greater value of $S_{co-occ}$ is, the more independent items a user has in their profile leading to high surprise score. In this work, $S_{co-occ}$ of a user profile is calculated by taking the mean value of the co-occurrence scores based on all items rated before a specific item in the user profile.

(3) **Coverage (Cov.)** measures the user's interest in the diversity of item categories. Following [37], we define *coverage* as the percentage of interacted categories in user profile:

$$\text{Coverage}(\mathcal{P}) = \frac{1}{|Cat|} \left| \bigcup_{i \in \mathcal{P}} Cat(i) \right| , \qquad (4)$$

where $Cat(i)$ and $|Cat|$ indicate set of categories covered by item $i$ and the total number of categories, respectively. Coverage ranges within $(0, 1]$, with 0 indicating coverage of no categories, and 1 of all categories.

(4) **Redundancy (Red.)** measures how many redundant categories the users have interacted with. Redundancy complements coverage, as coverage alone does not indicate the user's willingness to accept diverse recommendations [37]. Formally:

$$\text{Redundancy}(\mathcal{P}) = 1 - \frac{\left| \bigcup_{i \in \mathcal{P}} Cat(i) \right|}{\sum_{i \in \mathcal{P}} |Cat(i)|} , \qquad (5)$$

ranging within $[0, 1)$, where 0 indicates that no redundant item categories are in the user profile. The greater the value, the more redundant item categories are in the user profile.

## 3.2 Extracting user propensities

After defining the beyond-accuracy metrics of our interest, here we leverage the historical interactions and calculate beyond-accuracy metrics on the user profile based on the previously rated items. Our goal is to examine how much existing datasets support our hypotheses and how much users' historical interactions differ in terms of beyond-accuracy metrics.

To this aim, for each user, we calculate the beyond-accuracy metrics (i.e., features used for user clustering) mentioned in Sec. 3.1. Fig. 2 shows the min-max–normalized distribution of each value for all users in the MovieLens1M and Yelp datasets. The Yelp dataset shows more variation in user propensities. Among the analyzed metrics, co-occurrence-based surprise shows the highest variation in Yelp due to the high sparsity of the interaction matrix. In the MovieLens1M dataset, we see a higher variation among coverage and redundancy features indicating the role these metrics can play in characterizing users' expectations. The considerable variation of extracted features from user preferences highlights the potential benefit of clustering users. By grouping users with similar preferences, the underlying behavior and common expectations of these groups can be revealed. This increased understanding of user behavior can inform personalized recommendations that better meet their individual needs and preferences.

Therefore, we cluster users based on the extracted features with the $K$-means algorithm. To find the optimal number of clusters, we perform a preliminary experiment using the *Elbow* method alongside the *Silhouettes* score for different values of $K$. Our analysis of both datasets reveals the presence of four unique user groups with distinct behavior patterns. One cluster, for example, encompasses users who are highly active and have a preference for novel items, as observed in cluster 3 in the Yelp dataset. Another cluster consists of less active users who show interest in a wide range of categories, as seen in cluster 4 in the Yelp dataset (see Fig. 3). A greater number of clusters would result in clusters with smaller user sizes and little added value in minimizing inter-cluster variations. Table 1 shows the statistics of clustering results. We depict the center of each cluster in Fig. 3. There is a considerable distance between the cluster centroids. A closer look at the centroids in the MovieLens1M reveals some interesting results. As an example, users in cluster 4 are highly active users (with 479.57 average interactions) and are particularly interested in specific items with low coverage and redundancy and the highest novelty score. In contrast, clusters 2 and 3 contain inactive users (24.33 and 47.95 average interactions, respectively) with the difference that users in cluster 3 are interested in a narrower range of categories. The Yelp dataset seems to exhibit a similar pattern, suggesting that the $K$-means algorithm can cluster users to capture their behavioral differences, as intended.

## 3.3 Recommendation algorithm performance on calibration

Next, we intend to analyze how different recommendation algorithms, ranging from traditional to deep recommendation models, serve different user clusters in calibrating their expectations in the recommended lists. Particularly, we include three traditional methods (MF [23], BPR [31], and WMF [16]) and a deep recommendation model (VAECF [26]). In addition, we include a non-personalized

baseline method, MostPop (ranking items based on their global popularity), for further investigation and comparison. To this end, we extract the beyond-accuracy features from the recommendation lists generated by each recommendation algorithm. Based on a feature vector for each user in the train data (expected output) and the recommendation list (algorithm's output), we can calculate the distance between these two vectors as a measure of algorithm miscalibration. *Miscalibration* can be formally defined as the deviation of the algorithm's recommended items from the user's historical preferences in terms of beyond-accuracy metrics. Fig. 4 illustrates the miscalibration of algorithms regarding beyond-accuracy metrics using Euclidean distance[1] averaged over all users. The values of this figure illustrate the incapability of current recommendation algorithms in satisfying users' beyond-accuracy expectations. Generally, all models have higher miscalibration on the MovieLens1M dataset than Yelp. Among models, VAECF seems to be the best-performing in beyond-accuracy calibration, followed by MF and WMF, although this capability is domain-dependent. Moreover, as expected, the MostPop model provides users with the highest level of miscalibration by recommending the same popular items to all clusters. Hence, it is imperative to devise techniques to increase the system's engagement rate by providing personalized recommendations tailored to each user's taste.

**Table 1: Clusters statistics: $|\mathcal{U}|$ is the number of users in a cluster, $|\mathcal{T}|$ is the number of interactions, $\frac{|\mathcal{T}|}{|\mathcal{U}|}$ is the average number of interactions per user.**

| Dataset | Cluster | $|\mathcal{U}|$ | $|\mathcal{T}|$ | $\frac{|\mathcal{T}|}{|\mathcal{U}|}$ |
|---|---|---|---|---|
| | 1 | 2503 | 363157 | 145.09 |
| **MovieLens1M** | 2 | 1143 | 27815 | 24.33 |
| | 3 | 1715 | 82228 | 47.95 |
| | 4 | 679 | 325630 | 479.57 |
| | 1 | 608 | 18866 | 31.03 |
| **Yelp** | 2 | 534 | 30255 | 56.66 |
| | 3 | 120 | 17740 | 147.83 |
| | 4 | 905 | 21421 | 23.67 |

## 3.4 Summary

We summarize our findings as follows:

- **Answer to RQ1**: We calculate the beyond-accuracy metrics for each user. The distribution of these metrics shows significant variations among users in both MovieLens1M and Yelp, with Yelp showing more variations. We use the $K$-means algorithm to cluster users based on these metrics and find that there are four distinct groups of users. The clustering results reveal that users in different clusters have distinct preferences for beyond-accuracy metrics, and the $K$-means algorithm successfully separates them into clusters with meaningful differences. Such variations highlight the need for personalizing beyond-accuracy qualities accordingly.

---

[1]The results obtained with other distance metrics (e.g., L1 norm and cosine distance) are similar.

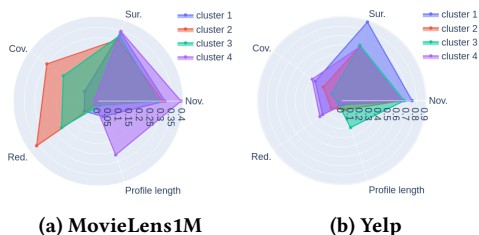

**(a) MovieLens1M**          **(b) Yelp**

**Figure 3: User clusters centers for beyond-accuracy metrics (best viewed in color).**

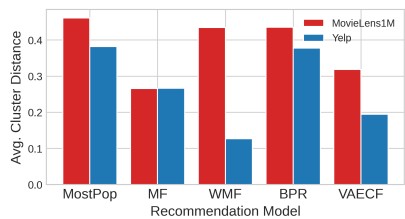

**Figure 4: Distance between user expectations and the provided recommendations.**

- **Answer to RQ2**: Based on the analysis of the miscalibration of recommendation algorithms in terms of beyond-accuracy metrics, it is clear that current models have a limited ability to satisfy individual variations in user preferences. This is evident in the high miscalibration values observed for all models, particularly on MovieLens1M. Our analysis shows that there is no user group that receives notably favorable recommendations in terms of beyond-accuracy calibration. Even among the most active users represented by clusters 1 and 4, the recommendations are highly miscalibrated. This highlights the importance of devising techniques that can effectively address individual variations in user preferences for beyond-accuracy metrics.

## 4 PERSONALIZED BEYOND-ACCURACY CALIBRATION ALGORITHM

### 4.1 Problem formulation

Let $\mathcal{U}$ and $\mathcal{I}$ be the set of users and items, respectively. We assume user $u \in \mathcal{U}$ is provided with a list of top-$S$ items, $L_u^S$, given a predicted relevance score matrix, $R^{|\mathcal{U}| \times S}$. As analyzed in the previous section, state-of-the-art recommender systems fail to generate beyond-accuracy-calibrated recommendations as they are accuracy-optimized. We provide a re-ranking framework to calibrate the baseline recommendation list, such that the new list aligns better with users' preferences considering both PBAC and accuracy objectives.

Let $\mathcal{M} = \{\mathbf{M_1}, \mathbf{M_2}, ...\mathbf{M_H}\}$ be a set of metrics that measures the quality of recommendation beyond accuracy. Let $\mathcal{P}_u$ be the set of items user $u$ has interacted with in the training data. To measure the miscalibration in a list over metric $\mathbf{M_k} \in \mathcal{M}$, we calculate the distance between the user's expected outcome extracted from the user profile, $c_u^k = \mathbf{M_k}(\mathcal{P}_u)$, and the recommended list, $o_u^k = \mathbf{M_k}(L_u^S)$,

**Algorithm 1:** The PBAC Greedy Polynomial Optimization

---

**Input** : $\mathcal{U}, \mathcal{I}, \lambda, \alpha, K$
**Output**: Item recommendation matrix $X^*$
/* $X^*$ is a $|\mathcal{U}| \times S$ binary matrix with $X_{u,i} = 1$ when the item $i$ is recommended to user $u$ and $X_{u,i} = 0$ otherwise. */
1 $R \leftarrow$ Run the baseline algorithm and store top-$S$ scores for each user
  /* $R$ is a $|\mathcal{U}| \times S$ matrix of relevance scores */
2 Solve linear relaxation of MILP problem 6 (i.e., with $0 \leq X_{ui} \leq 1$ instead of $X_{ui} \in \{0, 1\}$)
3 Round optimal $X_{ui}$ values to nearest integers
4 **for** $u \in \mathcal{U}$ **do**
5    **while** $\sum_{i=1}^{S} X_{ui} \neq K$ **do**
6      **if** $\sum_{i=1}^{S} X_{ui} > K$ **then**
7        $i' \leftarrow \underset{i}{\operatorname{argmin}} R_{ui} \quad \forall i \in \{i \mid X_{ui} = 0\}$
8        $X_{ui'} \leftarrow 0$
9      **else**
10        $i' \leftarrow \underset{i}{\operatorname{argmax}} R_{ui} \quad \forall i \in \{i \mid X_{ui} = 1\}$
11        $X_{ui'} \leftarrow 1$
12      **end**
13    **end**
14 **end**
15 **return** $X (\equiv X^*)$

---

using a distance function $\mathbf{d}(.)$, e.g., cosine distance or $L_p$ norm distance. We define the $\epsilon$-calibration objective as follows:

*Definition 4.1.* A recommender system satisfies $\epsilon$-calibration w.r.t. beyond-accuracy metric $\mathbf{M_k}$ if

$$G_k(L^S, \mathcal{U}) = \frac{1}{|\mathcal{U}|} \sum_{u \in \mathcal{U}} |\mathbf{d}(c_u^k, o_u^k)| \leq \epsilon \quad \forall u \in \mathcal{U}$$

where $L^S = \{L_u^S\}_{u \in \mathcal{U}}$. Also, it satisfies $\epsilon$-PBAC requirement when $G_k(L^S, \mathcal{U}) \leq \epsilon \quad \forall \mathbf{M_k} \in \mathcal{M}$

In other words, a recommender system satisfies the $\epsilon$-PBAC requirement if $G_k$ is less than or equal $\epsilon$ for all beyond-accuracy metrics, where $G_k$ is the average distance (over users) between the beyond-accuracy metrics of the recommended list and the user's profile. According to this definition, $\epsilon$ represents the degree of strictness imposed by PBAC; when epsilon is zero the optimal calibration is achieved. Note, nonetheless, excessive adherence to PBAC may result in a loss of accuracy. Therefore, we optimize for both PBAC and accuracy metrics.

### 4.2 Optimization model

This part outlines an algorithm that generates a calibrated list by properly selecting $K$ items for each user $u$ among its top-$S$ items, $L_u^S$, where $K \leq S \leq |\mathcal{I}|$. We define a binary decision matrix $X = [X_{ui}]_{|\mathcal{U}| \times S}$ to denote whether the item $i$ is recommended to user $u$ in the calibrated list. For each user $u$, we represent the beyond-accuracy calibrated top-$K$ list as a binary vector $X_u =$

$[X_{u1}, X_{u2}, ..., X_{uS}]$ when $\sum_{i=1}^{S} X_{ui} = K$. We formulate this calibration as an optimization problem to maximize the total sum of relevance scores constrained by $\epsilon$-calibration requirement as:

$$\max_{X} \quad \sum_{u \in \mathcal{U}} \sum_{i=1}^{S} R_{ui}.X_{ui}$$

$$\text{s.t.} \quad G_k(L^S(X), \mathcal{U}) \leq \epsilon \qquad \forall M_k \in \mathcal{M} \tag{6}$$

$$\sum_{i=1}^{S} X_{ui} = K \qquad X_{ui} \in \{0, 1\}$$

where $L^S(X)$ is the top-$S$ lists, $L^S$, induced from $X$. Any feasible solution of problem 6 recommends exactly $K$ items to each user and is guaranteed to satisfy $\epsilon$-PBAC requirement. The optimal solution has the highest sum of relevance to users possible under the constraints. It can be challenging to select the appropriate $\epsilon$ value since a small value leads to an empty solution space, while large values for $\epsilon$ have little impact on PBAC. Thus, we append the PBAC constraint to the optimization objective as a regularized penalty term with importance weight (hyper-parameter) $\lambda_k$ for each $M_k$. We also apply a weighting parameter $\alpha_u$ for each user $u$. We assume all the users are of the same importance and apply same $\alpha_u$ to maximize total relevance score[2]. In summary, we replace the objective function in probelm 6 with:

$$\sum_{u \in \mathcal{U}} \sum_{i=1}^{S} \alpha_u R_{ui}.X_{ui} - \sum_{j} \lambda_j.G_j(X, \mathcal{U}) \tag{7}$$

We note that the proposed optimization problem is a mixed-integer-linear programming (MILP) problem. Although MILP is an NP-hard problem, we can use a wide variety of effective heuristic solvers that provide a satisfactory and feasible solution in practice.[3] We also provide a greedy algorithm to make it suitable for the large-scale recommendation settings by reducing the MILP to a Linear Programming (LP) problem. This can be achieved by replacing the binary constraint $X_{ui} \in \{0, 1\}$ by $0 \leq X_{ui} \leq 1$ (Lines 1 and 2 in Algorithm 1) and rounding the optimal $X_{ui}$ values to nearest integer (Line 3). For each user $u$, while the total number of recommended items is not equal to $K$ (i.e., $\sum_{i=1}^{S} X_{ui} \neq K$), we iteratively add (remove) items with the highest (lowest) relevance scores (Lines 8 and 11). This iterative adding and removing naturally results in the desired $K$ total number of items recommended to each user, while only selecting the most relevant items. While our algorithm does not provide the global optimum, our experiments indicate that it can provide a near-optimal solution relatively quickly. The pseudo-code is presented in Algorithm 1.

## 5 EXPERIMENTAL SETUP

In this section, we describe our experiments. To ensure reproducibility, we used the open-source recommendation toolkit Cornac [33], and make our code open source upon acceptance.

**Datasets.** We use two public datasets, MovieLens1M and Yelp. We apply 20-core pre-processing on the datasets to make sure each user/item has sufficient feedback; i.e., each training user/item has at least 20 ratings. Table 2 shows the specs of the datasets, where $|Cat|$ is the number of categories in the dataset.

---

[2]We leave the impact of varying weights for different user groups to future work
[3]For instance, the Gurobi solver (https://www.gurobi.com).

**Table 2: Characteristics of datasets**

| Dataset | $|\mathcal{U}|$ | $|\mathcal{I}|$ | $|\mathcal{T}|$ | $|Cat|$ | %sparsity |
|---|---|---|---|---|---|
| Yelp | 7,135 | 16,621 | 1,137,521 | 325 | 95.05% |
| MovieLens1M | 6,040 | 3,260 | 998,538 | 18 | 94.93% |

**Evaluation.** We perform a 80/20 temporal split on each dataset for train and test data, respectively. For clustering, we extract the users' beyond-accuracy using the metrics introduced in Sec. 3 and perform a $K$-means clustering. We use $K = 4$ based on the elbow method and silhouette score. To evaluate the models' top-10 recommendation list, we employ well-known accuracy-based metrics such as nDCG, Precision, and Recall, as well as the beyond-accuracy metrics of Sec. 3.1. For the optimization experiment, we use the same regularization weight for all beyond-accuracy metrics, i.e., $\lambda_j = \lambda$ for all $j$, and re-ranking is performed on baseline top-50 ($S = 50$) recommendation list[4]. Since accuracy and beyond-accuracy metrics usually oppose each other [17], we define weighted average recommendation performance (ARP) metric to evaluate the overall performance of the models in terms of both types of metrics:

$$ARP(w) = (1 - w) \cdot \Delta(\%Dist.) + w \cdot \Delta(\%nDCG) \tag{8}$$

$\Delta(\%Dist.)$ is the percentage of improvement in miscalibration degree (see Definition 4.1) and $\Delta(\%nDCG)$ is the percentage of drop in nDCG metric after optimization. The weighted ARP represents the overall improvement of the recommendation system after optimization. We considers three scenarios for evaluation: higher priority for beyond-accuracy calibration (w=0.3), equal priority for both (w=0.5), and higher priority for accuracy (w=0.7). We tune $\lambda$ as a hyperparameter to maximizes $ARP(0.5)$ on train data. The results of an ablation study on various $\lambda$ is in Sec. 6.

**Baselines.** We compare our approach to various traditional and modern deep learning models, as suggested by Dacrema et al. [10] including:

- **MostPop**: The most popular items are recommended to users using a simple and non-personalized method. The measure of popularity is the number of interactions.
- **MF** [23]: This is an ordinary matrix factorization which decomposes the user-item interaction matrix into lower dimension rectangular matrices.
- **WMF** [16]: This matrix factorization method assumes latent features are independent for a pair of items and assigns smaller weights to negative samples.
- **BPR** [31]: BPR instantiates a zero mean Gaussian prior based on the latent factors of users and items interaction matrix and adds it to the supervised loss as an $L_2$ regularization term.
- **VAECF** [26]: This method uses variational autoencoders with multinomial Bayesian inference to estimate the parameters of a generative model.

We implement our algorithm using these baselines to show how it achieves the desirable performance on calibration metrics in overall recommendation performance.

---

[4]Our preliminary experiments show an increase in $S$ enhances the model's performance, however, this comes with a much higher computational cost. $S = 50$ appears to be at the balance point of this trade-off.

**Table 3: Performance of the baselines (N) and our proposed algorithm on the Yelp dataset. The evaluation metrics are calculated based on the top-10 predictions in the test set. The gray highlight indicates the improvement in miscalibration measured by Dist. metric compared to N.**

| Cluster | Model | Type | Beyond-Accuracy Metrics | | | | | Accuracy Metrics | | | ARP(w) | | |
|---|---|---|---|---|---|---|---|---|---|---|---|---|---|
| | | | Nov. | Red. | Cov. | Sur. | Dist. ↓ | Precision | Recall | nDCG | ARP(0.3) | ARP(0.5) | ARP(0.7) |
| $C_1$ | MostPop | N | 0.0 | 0.614 | 0.6344 | 0.5543 | 0.9428 | 0.0005 | 0.0005 | 0.0004 | 0 | 0 | 0 |
| | | PBAC | 0.0 | 0.6436 | 0.5682 | 0.5666 | ↑1.0694% **0.9327** | 0.0005 | 0.0005 | 0.0004 | 0.7486 | 0.5347 | 0.3208 |
| | MF | N | 0.9281 | 0.2693 | 0.7673 | 0.0517 | 0.9511 | 0.0005 | 0.0006 | 0.0004 | 0 | 0 | 0 |
| | | PBAC | 0.8489 | 0.4782 | 0.5648 | 0.1006 | ↑10.4032% **0.8521** | **0.0013** | **0.0011** | **0.0009** | 43.1101 | 64.9147 | 86.7192 |
| | WMF | N | 0.6119 | 0.5381 | 0.5115 | 0.8715 | 0.3548 | 0.0387 | 0.0552 | 0.0392 | 0 | 0 | 0 |
| | | PBAC | 0.6215 | 0.5368 | 0.5117 | 0.8682 | ↑1.2637% **0.3503** | **0.0396** | **0.0579** | **0.0394** | 1.0299 | 0.8741 | 0.7183 |
| | BPR | N | 0.0 | 0.6437 | 0.5679 | 0.5676 | 0.9324 | 0.0005 | 0.0005 | 0.0004 | 0 | 0 | 0 |
| | | PBAC | 0.0121 | 0.6089 | 0.4955 | 0.5754 | ↑4.4049% **0.8913** | 0.0005 | 0.0005 | 0.0004 | 3.0834 | 2.2024 | 1.3215 |
| | VAECF | N | 0.4144 | 0.5659 | 0.4821 | 0.9411 | 0.481 | **0.0627** | **0.084** | **0.0657** | 0 | 0 | 0 |
| | | PBAC | 0.5941 | 0.4946 | 0.553 | 0.8809 | ↑27.0283% **0.351** | 0.0398 | 0.0532 | 0.0421 | 8.1607 | -4.4177 | -16.9961 |
| $C_2$ | MostPop | N | 0.0 | 0.614 | 0.6344 | 0.5543 | 0.8902 | 0.0427 | 0.0318 | 0.0458 | 0 | 0 | 0 |
| | | PBAC | 0.0 | 0.6437 | 0.568 | 0.5666 | ↑1.217% **0.8793** | **0.0431** | **0.0322** | **0.046** | 1.0013 | 0.8575 | 0.7137 |
| | MF | N | 0.9281 | 0.2693 | 0.7673 | 0.0517 | 0.8207 | 0.0047 | 0.0037 | 0.0049 | 0 | 0 | 0 |
| | | PBAC | 0.8356 | 0.4883 | 0.5607 | 0.097 | ↑12.6833% **0.7166** | **0.0067** | **0.0055** | **0.006** | 15.4257 | 17.254 | 19.0823 |
| | WMF | N | 0.4473 | 0.5794 | 0.5761 | 0.5453 | 0.5863 | **0.0433** | **0.033** | **0.0452** | 0 | 0 | 0 |
| | | PBAC | 0.4673 | 0.5644 | 0.5619 | 0.5415 | ↑4.3288% **0.5609** | 0.0412 | 0.0307 | 0.0444 | 2.5079 | 1.2939 | 0.0799 |
| | BPR | N | 0.0 | 0.6437 | 0.5679 | 0.5676 | 0.8793 | 0.0431 | 0.0322 | **0.0472** | 0 | 0 | 0 |
| | | PBAC | 0.0121 | 0.6076 | 0.4969 | 0.5748 | ↑6.1119% **0.8255** | **0.0433** | **0.0323** | 0.0462 | 3.6015 | 1.928 | 0.2544 |
| | VAECF | N | 0.2202 | 0.5811 | 0.5862 | 0.6373 | 0.7006 | **0.0594** | **0.0451** | **0.0607** | 0 | 0 | 0 |
| | | PBAC | 0.4421 | 0.534 | 0.5542 | 0.6062 | ↑23.0442% **0.5391** | 0.0358 | 0.0261 | 0.0386 | 5.1887 | -6.7149 | -18.6186 |
| $C_3$ | MostPop | N | 0.0 | 0.614 | 0.6344 | 0.5543 | 1.0117 | 0.0783 | 0.0233 | 0.0786 | 0 | 0 | 0 |
| | | PBAC | 0.0 | 0.6435 | 0.5684 | 0.5666 | ↑1.4443% **0.9971** | **0.085** | **0.025** | **0.0826** | 2.5593 | 3.3027 | 4.0461 |
| | MF | N | 0.9281 | 0.2693 | 0.7673 | 0.0522 | 0.921 | 0.0125 | 0.0031 | 0.0115 | 0 | 0 | 0 |
| | | PBAC | 0.8114 | 0.5055 | 0.5612 | 0.0905 | ↑9.2198% **0.8361** | **0.0158** | **0.0041** | **0.0136** | 11.8476 | 13.5995 | 15.3514 |
| | WMF | N | 0.417 | 0.5809 | 0.5917 | 0.5718 | 0.748 | **0.0675** | **0.0201** | **0.0611** | 0 | 0 | 0 |
| | | PBAC | 0.4435 | 0.5584 | 0.5623 | 0.5691 | ↑5.6991% **0.7054** | 0.0625 | 0.0185 | 0.0585 | 2.6922 | 0.6875 | -1.3171 |
| | BPR | N | 0.0 | 0.6437 | 0.5679 | 0.5675 | 0.9969 | 0.085 | **0.025** | **0.0847** | 0 | 0 | 0 |
| | | PBAC | 0.0116 | 0.6134 | 0.4939 | 0.5753 | ↑5.71% **0.94** | **0.0858** | 0.0249 | 0.0832 | 3.436 | 1.92 | 0.404 |
| | VAECF | N | 0.2274 | 0.5799 | 0.5737 | 0.6639 | 0.8199 | **0.0975** | **0.029** | **0.0946** | 0 | 0 | 0 |
| | | PBAC | 0.4717 | 0.5448 | 0.5319 | 0.6504 | ↑18.7358% **0.6663** | 0.075 | 0.0223 | 0.0772 | 7.6089 | 0.191 | -7.227 |
| $C_4$ | MostPop | N | 0.0 | 0.614 | 0.6344 | 0.5543 | 0.7594 | **0.021** | **0.0358** | **0.0214** | 0 | 0 | 0 |
| | | PBAC | 0.0 | 0.6435 | 0.5682 | 0.5666 | ↑0.7577% **0.7536** | 0.0209 | 0.0349 | 0.0213 | 0.4358 | 0.2213 | 0.0067 |
| | MF | N | 0.9281 | 0.2693 | 0.7673 | 0.0517 | 0.7256 | 0.0015 | 0.0019 | 0.0016 | 0 | 0 | 0 |
| | | PBAC | 0.8436 | 0.4862 | 0.5672 | 0.1057 | ↑17.7504% **0.5968** | 0.0015 | **0.002** | 0.0016 | 13.2879 | 10.3128 | 7.3378 |
| | WMF | N | 0.5415 | 0.5526 | 0.569 | 0.5516 | 0.335 | **0.0206** | **0.0345** | **0.021** | 0 | 0 | 0 |
| | | PBAC | 0.545 | 0.5482 | 0.5622 | 0.5505 | ↑2.3713% **0.3271** | 0.0198 | 0.0331 | 0.0203 | 0.7269 | -0.3693 | -1.4655 |
| | BPR | N | 0.0 | 0.6437 | 0.5679 | 0.5676 | 0.7536 | **0.0209** | **0.0349** | **0.0213** | 0 | 0 | 0 |
| | | PBAC | 0.0117 | 0.6072 | 0.5006 | 0.5743 | ↑5.3275% **0.7134** | 0.0173 | 0.03 | 0.0188 | 0.2858 | -3.0753 | -6.4365 |
| | VAECF | N | 0.22 | 0.5958 | 0.5779 | 0.6468 | 0.5541 | **0.0345** | **0.0556** | **0.0366** | 0 | 0 | 0 |
| | | PBAC | 0.4204 | 0.5395 | 0.55 | 0.6032 | ↑33.5595% **0.3681** | 0.02 | 0.0319 | 0.0214 | 11.0346 | -3.982 | -18.9986 |

*Nov.: Global Long-tail Novelty, Red.: Redundancy, Cov.: Coverage, Sur.: Co-occurrence-based Surprise, Dist.: Euclidean Distance*

**Evaluation settings.** We adopt the baseline algorithms with the default parameter settings of their papers with embedding size of 50 for all baseline algorithms. We set the learning rate to 0.001. Early stopping is applied and the best models are selected based on cross-validation. The model parameters are updated using Adam [21] as the optimization algorithm.

## 6 RESULTS AND DISCUSSION

In this section, notation N is used to represent the original baseline top-10 recommendation while PBAC refers to our optimized re-ranking method.

**Improving beyond-accuracy.** Fig. 5 presents the average miscalibration across all beyond-accuracy metrics for the users in various clusters, both before and after optimization (see Definition 4.1). The results of the baseline demonstrate that conventional collaborative filtering algorithms are not effective for beyond-accuracy calibration (as mentioned in Sec. 3). Our proposed algorithm improves (reduces) the miscalibration of recommendations for all users in the Yelp dataset, as in Fig. 5b. The degree of improvement varies across models and clusters, with the highest improvement observed in the VAECF method and Cluster 4. The Dist. column in Table 3 shows average improvement on miscalibration for Yelp across all baselines. A lower distance indicates less miscalibration. The reduction of miscalibration degree among clusters ($C_1, C_2, C_3, C_4$) are (8.83%, 9.47%, 8.16%, 11.95%), demonstrating the effectiveness of our PBAC algorithm in addressing varying user preferences. Comparing the

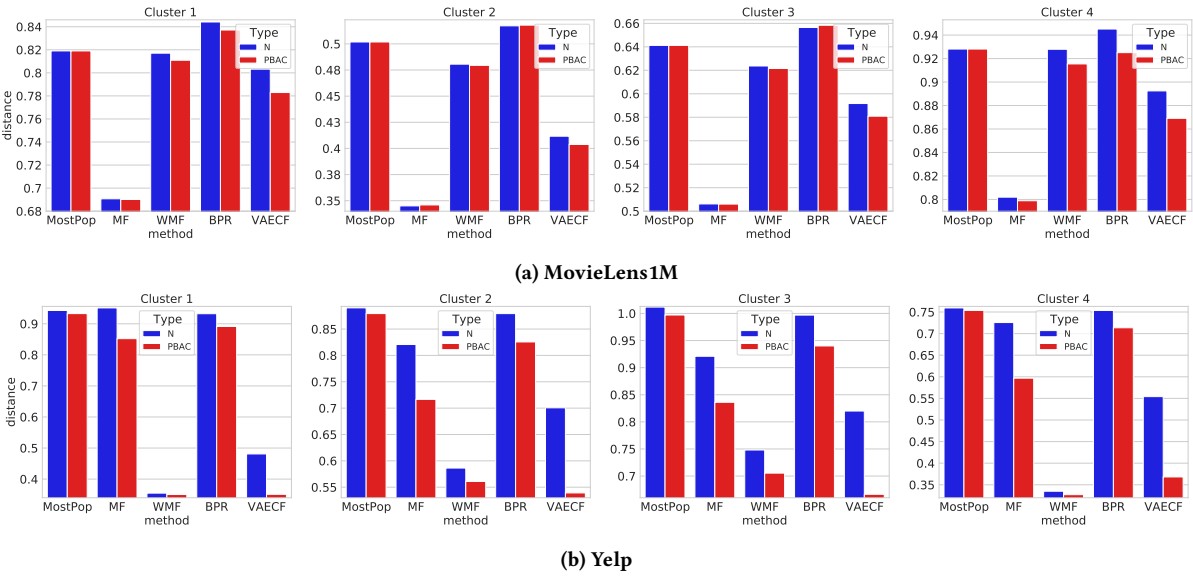

(a) MovieLens1M

(b) Yelp

**Figure 5: Distance between train and recommendation list cluster centers before and after optimization. N refers to the baseline performance before beyond-accuracy calibration.**

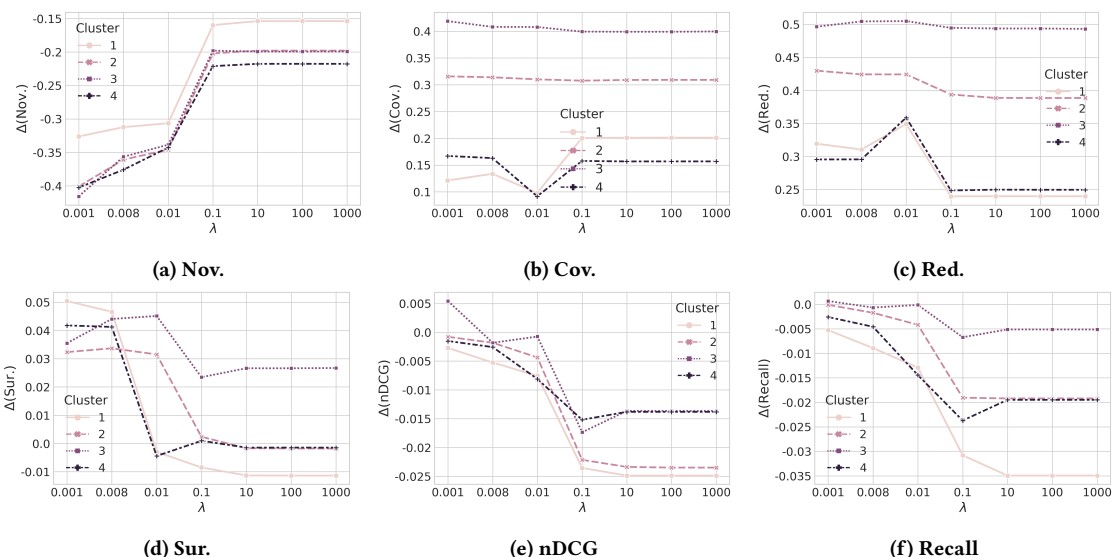

(a) Nov.                              (b) Cov.                              (c) Red.

(d) Sur.                              (e) nDCG                              (f) Recall

**Figure 6: The change of accuracy and beyond-accuracy metrics with PBAC with respect to the $\lambda$ parameter.**

datasets, it can be observed that the Yelp dataset exhibits less variation among clusters for reducing miscalibration, with the PBAC algorithm performing relatively consistent across all clusters.

The analysis of PBAC algorithm on different clusters in Movie-Lens1M reveals a higher degree of variation, with some clusters experiencing only slight improvement, while others show substantial improvement. Notably, Fig. 5a, highlights that cluster $C_4$ has a significant reduction in miscalibration compared to other clusters. This variation in improvement is attributed to the preferences of users in that cluster. It can be seen in Fig. 3a that the center

of beyond-accuracy metrics of cluster $C_4$ has a higher distance to other clusters, indicating a greater degree of variance in user tastes. This is due to underlying data characteristics, as Yelp has higher sparsity but more balanced interactions (see [41]). In contrast, the MovieLens1M dataset is heavily influenced by popularity, with certain items receiving the majority of ratings. This popularity bias, as described in [3], limits the ability of baseline algorithms to recommend novel and unexpected items to satisfy the diverse tastes

of users. As a result, the effectiveness of the PBAC algorithm in reducing miscalibration may be limited compared to its performance on the Yelp dataset.

Next, we investigate whether our proposed PBAC algorithm improves the overall performance of recommendation with respect to accuracy and beyond-accuracy calibration simultaneously, i.e., whether our algorithm is capable of balancing multiple objectives optimally. This can be addressed by observing the patterns in $ARP(w)$ columns in Table 3, which are the weighted average between the percentage of improvement in miscalibration and the changes in nDCG values after optimization (formally defined at Eq. (8)); hence positive values indicate an overall improvement. When assigning higher importance to beyond-accuracy calibration in comparison to accuracy (ARP(0.3) column), all PBAC models improve the overall recommendation performance. In the case of equality of importance between the two objectives (ARP(0.5) column), the PBAC model outperforms the baselines in clusters $C_1$, $C_2$, and $C_3$. As baseline models are already optimized for accuracy, one may expect PBAC models to perform worse than baselines when a higher weight is assigned to the accuracy objective (ARP(0.7) column). Surprisingly, in many cases, PBAC models outperform the accuracy-optimized baseline models as can be seen in the $ARP(0.7)$ column in Table 3, e.g., the MF model in cluster $C_1$ with the value of 86.72. Further, the average $ARP(w)$ values for weights (0.3, 0.5, 0.7) is (6.8886, 5.0767, 3.2648) indicating an average improvement over baselines in all three scenarios. Hence, our proposed PBAC algorithm effectively balances the trade-off between accuracy and beyond-accuracy calibration in recommender systems for various user clusters. This addresses **RQ3** and demonstrates the robustness and versatility of the PBAC model in optimizing multiple objectives in a real-world setting.

**Baseline recommender model dependency.** The underlying recommendation algorithm has a significant impact on the effectiveness of the proposed PBAC re-ranking model. Because baseline recommendation algorithms are inputs to PBAC re-ranking models, so their performance is highly correlated with their effectiveness. Investigating the reduction in miscalibration degree, Fig. 5b illustrate that in Yelp PBAC performs considerably better on the VAECF baseline, followed by MF and BPR, with an average reduction of (25.59%, 12.51%, and 5.38%) among the four clusters. Likewise, the VAECF model exhibits a more substantial average miscalibration reduction than other methods on the MovieLens1M dataset, as shown in Fig. 5a. Essentially this shows that VAECF is learning the underlying user-item representation more accurately and in the most unbiased way. This finding confirms the results reported in [28] on the predictive ability of the VAECF model for beyond-accuracy metrics. On the other hand, among all clusters and datasets, the MostPop algorithm shows the lowest reduction in miscalibration with an average improvement of 1.12%. This was expected as this algorithm is non-personalized, and the items recommended by the baseline recommendation list are similar for all users. Thus, calibrating over parameters such as novelty and surprise for users with different tastes may be less effective since there is not enough variety in recommended items. Comparing the average recommendation performance (ARP), as can be seen in Table 3, the VAECF

model improves beyond accuracy calibration significantly; however, it sacrifices accuracy in the process. This results in negative ARP when higher importance is assigned to accuracy objective, i.e., ARP(0.7). Considering equal importance at ARP(0.5), we see positive values for all models except VAECF in cluster $C_1$, $C_2$, and $C_3$. In addition, when beyond-accuracy calibration is prioritized, ARP(0.3), we can see positive average ARP values among all clusters and baseline algorithms as expected from the PBAC model performance (see Table Table 3). This result addresses **RQ4** and shows that collaborative filtering algorithms may have difficulty learning user preferences and assigning high relevance scores to items that appear to be relevant to users from both accuracy and beyond-accuracy perspectives simultaneously. Consequently, the effectiveness of the PBAC re-ranking model is affected by the recommendation algorithm and is often associated with trade-offs between accuracy and beyond-accuracy calibration

**Ablation study.** We analyze the impact of the optimization regularization parameter $\lambda$ from Eq. (7). The larger values of $\lambda$ are expected to produce recommendations calibrated to a user's taste in beyond-accuracy metrics. However, the excessive pursuit of calibration is unnecessary and could have an adverse impact on the overall recommendation performance. Therefore, we are interested in studying how different values of $\lambda$ in Eq. (7) can influence the overall performance of the recommender system, i.e., total accuracy (nDCG, Recall), and beyond-accuracy metrics defined in Sec. 3.

As indicated in Fig. 6, $\lambda$ exhibits a more "accuracy-centric" behavior, indicating that this parameter affects overall system accuracy more significantly than beyond-accuracy metrics, especially with respect to coverage and redundancy. The reason why an excessively high value of $\lambda$ does not benefit us in the overall performance (the combination of beyond-accuracy calibration and accuracy objective), can be explained by the fact that many long-tail and surprising items lack sufficient preference scores (interactions), making it unclear whether including these items in recommendation lists will actually result in user appreciation. Thus, choosing the appropriate model parameters is essential to increasing the level of personalization and overall utility of the marketplace.

## 7 CONCLUSION AND FUTURE WORK

This paper addresses the beyond-accuracy metrics calibration from the users' perspective. We demonstrated that users have eclectic preferences when it comes to beyond accuracy qualities such as diversity and novelty. We further showed that recommender systems designed to maximize accuracy fail to calibrate the generated recommendations with respect to beyond-accuracy metrics. We address this issue by utilizing user profile interactions and an efficient re-ranking algorithm; the calibrated recommendations should match user expectations for all clusters of users within a range of tolerance. Extensive experiments confirm that our method can achieve a desirable point in the trade-off between ranking accuracy and beyond-accuracy calibration.

For future work, we plan to extend our analysis to multi stakeholder settings, including producers or side-stakeholders, as an additional objective in the optimization framework. We can also analyze personalized beyond-accuracy calibrations in online settings as users' tastes change dynamically over time.

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
