# OpenReview forum: "Personalized Beyond-accuracy Calibration in Recommendation"
_ACM.org/SIGIR/ICTIR/2024/Conference — ICTIR 2024_

### Official Review · Reviewer_EVxN · 2024-05-14

**Rating:** 1
**Confidence:** 4

**Objective Part Of Review:**

The paper proposes the idea that recommender systems should cover different competing aspects (accuracy and beyond-accuracy measures) proportional (i.e. calibrated) to user needs. Whilst there are previous works on combining accuracy and beyond-accuracy measures (e.g. "Personalizing recommendation diversity based on user personality" by Wu et al.) as well as calibrating specific item groups (e.g. "A Constrained Optimization Approach for Calibrated Recommendations" by Seymen et al.), to my knowledge calibrating accuracy and beyond-accuracy metrics to user history is a novel idea.

The authors introduce concepts and define notation clearly. The abstract and the introduction provide a good overview of what the paper is trying to tackle and its connection and differences with traditional calibration in RecSys.

The authors provide a reasonable overview of related directions and works. The authors then highlight relevant beyond-accuracy metrics carry out user segmentation, identifying user groups with widely different beyond-accuracy behaviour. It would have helped me to know which other metrics were considered and what were the correlations between different metrics to better understand the difficulty of balancing multiple beyond-accuracy metrics. Assuming the beyond-accuracy metrics are sufficiently different, the user segmentation analysis forms the basis of the rest of the work and makes sense. Here, and overall, author claims are generally supported by the experiments and figures.

Whilst the authors performed a 80-20 split, at least in the case of MovieLens, where temporal dependencies are known to be weak (e.g. "Widespread Flaws in Offline Evaluation of Recommender Systems" by Hidasi and Czapp), random non-time based cross validation could have been considered. This would have allowed the authors to provide variance/confidence intervals for their result table. This is particularly relevant because the setting explored by this paper is already inherently a static one - all metrics in the last 20% have to be calibrated with the first 80%, which ignores any temporal drift and the fact that there generally might be differences in how the user interacted with the platform in their first week on the platform and the most recent one.

The research questions make sense and the experiments are generally appropriately set up to answer them. There is one significant omission, which is the lack of any baselines that incorporate beyond-accuracy metrics. In particular, if beyond-accuracy metric correlation is still rather strong, then any standard diversity approach would also provide significant gains over the current baselines. And if the metrics are weakly correlated and standard beyond accuracy baselines struggle with calibration, it would still substantially help the reader appreciate the gap in the field. The authors should consider adding such a baseline for the camera-ready version or in their future work, whichever is applicable.

**Subjective Part Of Review:**

The paper is easy to read. The idea is (to my knowledge) novel and interesting. The clustering analysis is interesting and provides a novel perspective on a widely studied dataset. There is potential in developing the work further, in particular in relation to temporal drift in accuracy and beyond-accuracy metrics. There is also a sufficient connection with theory - the paper is good fit for this conference.

---

### Official Review · Reviewer_Yhrb · 2024-05-17

**Rating:** 1
**Confidence:** 3

**Objective Part Of Review:**

The authors consider a what if questions: what if users of a recommender system fell into distinct categories that had different expectations of the system in terms of getting outcomes, with some people perhaps wanting to be given more diversity than the average user, and others wanting more long-tail items, and others wanting more repeated/redundant items, and so on.  Could a system be constructed that catered to these various expectations, without trading away too much accuracy?

Then to answer that question, the authors follow a number of steps: (a) build clusters over user behavior, to tease out whether there are in fact different types of user; (b) consider how to measure possibly different blends of user goals; (c) propose a mechanism for reranking system output to balance different goals; and then (d) measure for each cluster of users how they would be affected by that heuristic in terms of a blend between sheer accuracy, and their secondary
preferences.

That's quite a complex tale, and I think there is enough here to warrant acceptance of the paper.  Probably no-one will agree with every assumption made at every step of the journey, but even if we regard the paper to be a sketch of what the itinerary might include, it makes for thoughtful reading.

**Subjective Part Of Review:**

The paper is rather complex, and I got lost in a few places.  I *think* I have understood it correct (viz my summary above), but an initial schematic that introduced that flow would have been nice.  So I encourage you to think carefully about the story-line, and how you are presenting it.

Why is Definition 4.1 in a smaller font than the rest of the text?
And in deed, why is there a single thing given this formal treatment?

In Table 3, is there a "bottom line" summary possible wrt to net effect for the overall set of users, weighting the four clusters by their size to get a "population calculation" outcome?

Why didn't you suppress all the zeros in every second position in the last col of Table 3?

Are the discovered clusters in the two parts of Figure 3 reflecting similar properties?  Looks like not?  Can you give a verbal summary of what type of user is reflected in each of the eight clusters that are depicted, and are there then any similarities between any elements of the two sets of discovered user clusters?  (And, could Figure 3 possibly be made any smaller??)

"Note, nonetheless" in Section 4 is at least one word too long.

I didn't understand what I was to infer from Figure 5, and it needs more explanation/interpretation to take the reader through it.

Check Eqn 5, it didn't feel right to be summing up the sizes of the clusters that the items are members of in the denominator.

Why is there a midline in the top part of Table 1?

Why are the values in table 3 expressed to different numbers of digits?  I understand why the columns would be different, but would normally expect that within a column all would have the same level of accuracy.  (Indeed, when I see different numbers of decimal places in data tables, I start fearing cut-paste errors of various sorts, which is definitely a worry in a research paper.)

---

### Official Review · Reviewer_iWa7 · 2024-05-17

**Rating:** 1
**Confidence:** 3

**Objective Part Of Review:**

The paper presents a re-ranking approach for recommender systems based on Beyond Accuracy metrics. The methodology used post-hoc is interesting and based on robust theoretical formulations. The experimentation is carried out on only a few no-personalized and customized techniques and therefore not very robust. Novelty is limited being similar approaches known in the literature. It would also be useful to release the sources to ensure replicability of the approach. Overall, the contribution is valid with medium robustness.

**Subjective Part Of Review:**

The research topic is interesting and relevant to the scientific community. It would be appropriate to extend the experiments conducted and provide a formal and practical pipeline to follow to adapt one's recommendation algorithm to the methodology proposed here.

---

### Official Review · Reviewer_rNwv · 2024-05-20

**Rating:** -1
**Confidence:** 4

**Objective Part Of Review:**

Recommender systems typically focus on optimizing accuracy in a supervised setting, at the same time, this often overlooks important aspects like diversity, novelty, and serendipity. Previous research has mainly addressed these issues from the provider's perspective, such as enhancing overall diversity to counteract popularity bias. This study highlights that users have distinct preferences for beyond-accuracy metrics. The authors hypothesize that users possess an implicit behavioural model that values more than just accuracy. They argue that recommender systems should reflect these user interests to provide a more comprehensive understanding of user behaviour and preferences, leading to more personalized recommendations. To achieve this, they propose a post-ranking greedy optimization algorithm that balances accuracy with user-centric beyond-accuracy preferences. Extensive experiments show that this method effectively manages the trade-off between ranking accuracy and these additional user preferences.

This work is interesting and timely.
Experiments on two real-world datasets demonstrate that users significantly differ in their preferences regarding beyond-accuracy metrics.

the contributions include  demonstrating the importance of beyond-accuracy calibration for users by analyzing their preferences, thus making a case for personalised retrieval.

**Subjective Part Of Review:**

Figure 2 is incomplete

"Our goal is to examine how much existing datasets support our hypotheses" - no explicit hypotheses is given and second-guessing authors is difficult

" we cluster users based on the extracted features with the 𝐾-means algorithm.  " - no description of features given and if you meant the parameters in section 3.1, then say so.

Table 1 - data in this table is given in an ad-hoc fashion. what does this mean to say 1 cluster? is this the entire collection?- Please explain.

"Figure 4: Distance between user expectations and the provided recommendations." - Please explian this figure properly.

 state of the art recommenders  ignores personalised systems - for example, https://arxiv.org/pdf/1706.04148

PBAC is a post-optimization method whereas current personalisation systems are end-to-end

---

### Meta-Review · Area_Chair_h93j · 2024-05-30

**Recommendation:** Accept (Oral)
**Confidence:** 3

**Metareview:**

The paper presents a re-ranking approach for recommender systems that emphasize beyond-accuracy metrics, such as diversity, novelty, and serendipity, calibrated to user-specific preferences. The authors propose a post-ranking greedy optimization algorithm to balance accuracy with user preferences. The study is based on the hypothesis that users have distinct, implicit behavioral models that value different metrics in recommendation systems. Experiments are conducted on two real-world datasets.

The major concerns on this paper is its clarity and completeness. Both reviewer rNwv and Yhrb find multiple places where the authors need to clarify more in order for readers to understand the paper clearly, including several problems on tables, figures, and definitions (see the detailed reviews). Yet, while the reviewers have different opinions on the reliability and significance of this paper’s contributions, they agree that the overall merits is above the threshold, so we recommend an accept at the end. If this paper is accepted, we would strongly suggest the authors to revise the paper and address the reviewers’ questions and concerns before submitting the final version.